# Nesting of groups: Links among personal magnetism, community trust, inequality perception and depression

**Peicheng Wang[1], Yujie Zhang[2]\***

**1** School of International and Public Affairs, Shanghai Jiao Tong University, Xuhui District, Shanghai, China, **2** School of Public Policy and Administration, Chongqing University, Shapingba District, Chongqing, China

\* zhangyujie@cqu.edu.cn, zhangyujie33@foxmail.com

## Abstract

### Background

In the context of China's rapid transition from a collectivist work-unit system to a more individualized, community-oriented society, this study investigates the relationship between personal magnetism—an individual's social appeal—and its protective role in mitigating depression.

### Objective

This study aims to examine the moderating role of perceived inequality and the mediating effect of community trust in the relationship between personal magnetism and depression.

### Methods

Utilizing longitudinal data from the China Family Panel Studies (CFPS) from 2018 to 2020, involving 26,711 respondents aged 16 to 96, this study employs multivariate regression models to explore these relationships.

### Results

The findings reveal that perceived inequality amplifies the negative relationship between personal magnetism and depression, suggesting that personal magnetism becomes less effective in alleviating depressive symptoms in contexts of higher inequality. Additionally, community trust is identified as a key mediator, explaining how strong social networks can buffer the psychological effects of inequality and enhance mental well-being.

### Conclusion

The findings underscore the importance of fostering community trust and addressing income inequality in public policy. By leveraging the positive effects of personal magnetism and social solidarity, targeted interventions can reduce depression and enhance collective well-being in societies undergoing socio-economic transformation.

**Data availability statement:** The minimal data set necessary to replicate the study's findings

has been uploaded to a stable, public repository. Readers can access the data at the following DOI: https://doi.org/10.7910/DVN/KFYAUD, hosted on the Harvard Dataverse (Version 1).

**Funding:** This study was supported by Chongqing Municipal Social Science Planning Doctoral and Cultivation Project [Grant 2023BS080]. The funder had no role in study design, data collection and analysis, decision to publish, or preparation of the manuscript.

**Competing interests:** The authors have declared that no competing interests exist.

## Introduction

China's accession to the World Trade Organization (WTO) marked the onset of an unprecedented economic expansion, catalyzing a profound restructuring of its social fabric [1]. This rapid economic transformation has deepened socioeconomic stratification, eroding communal networks that historically functioned as pillars of social trust and collective identity. As traditional support systems weaken, individuals face diminished access to trust-based social capital, contributing to a fragmentation of interpersonal relationships and a decline in community cohesion [2]. These structural shifts have significant mental health implications, as evidenced by the 6.8% lifetime prevalence of depressive disorders among Chinese adults, alongside persistently low treatment engagement rates, highlighting the urgency of addressing the psychosocial consequences of rapid societal change [3].

By examining the interplay between personal magnetism, community trust, inequality perception, and depression, this study provides critical insights into the mechanisms through which individuals navigate social disembedding and adapt to transforming trust structures. More broadly, this study advances an empirically grounded framework for understanding the role of social trust and interpersonal networks in fostering psychological resilience. Its findings offer policy-relevant strategies for rebuilding social cohesion, reducing mental health disparities, and strengthening collective well-being in societies undergoing profound structural transformation.

### Literature review

**Personal magnetism and depression.** The construct of personal magnetism occupies a critical position in psychological and social theory, particularly in its role as a protective factor against depressive symptoms. Grounded in social psychology, personal magnetism is conceptualized as an individual's capacity to attract and maintain social relationships, fostering strong interpersonal connections and social integration. These connections extend beyond superficial interactions, serving as psychosocial resources that reinforce self-worth, provide emotional support, and enhance resilience against adversity [4,5].

Empirical research has substantiated the protective effects of personal magnetism on mental health, emphasizing the pivotal role of social bonds in mitigating psychological distress. For instance, Kawachi and Berkman assert that the breadth and depth of an individual's social networks—core components of personal magnetism—buffer against stressors that contribute to depression [6]. Similarly, Cacioppo and Hawkley identify loneliness as the antithesis of social magnetism, demonstrating its strong association with depressive symptomatology, thereby underscoring the importance of social connectedness in mental health regulation [7].

In transitional societies such as China, where rapid socio-economic changes have eroded traditional social structures, personal magnetism assumes an even greater adaptive function. The dissolution of the work-unit (Danwei) system, which historically provided institutionalized social stability through workplace-based communities, has resulted in a fragmentation of trust networks, creating a void that personal magnetism can effectively bridge [8,9]. This structural transformation necessitates new mechanisms of social integration, positioning personal magnetism as a critical asset in the reconstruction of interpersonal networks and the preservation of psychological well-being [10].

Given that individuals with greater personal magnetism are more likely to establish and sustain meaningful social relationships, which in turn function as buffers against psychological distress [11,12], this study hypothesizes:

**Hypothesis 1:** Personal magnetism is negatively correlated with depression.

**The moderating role of perceived inequality.** The relationship between personal magnetism and depression is not uniform across social contexts but is contingent upon individuals' perceptions of income inequality. Perceived inequality serves as a moderator, shaping how personal magnetism influences mental health by altering individuals' social comparisons, expectations of fairness, and access to psychosocial resources. The moderating role of perceived inequality can be understood through the lens of inequity aversion theory, which suggests that individuals who perceive greater economic disparities may experience heightened psychological distress, as such perceptions often generate feelings of social injustice and status anxiety [13,14]. In this context, even individuals with high personal magnetism may struggle to leverage their social networks as effective protective mechanisms, as widespread perceptions of inequality can erode trust, intensify competition for resources, and amplify the psychological burden associated with social standing [15,16].

Furthermore, relative deprivation theory provides additional insight into how perceived inequality exacerbates the relationship between personal magnetism and depression. Relative deprivation occurs when individuals feel disadvantaged relative to their peers, fostering frustration, alienation, and lower psychological well-being [17,18]. Individuals with high personal magnetism often engage in frequent social comparisons, as their social standing is tied to peer validation and interpersonal influence. However, in environments characterized by high perceived inequality, upward social comparisons may become more distressing, leading to diminished self-worth and increased vulnerability to depressive symptoms. In contrast, in societies with lower perceived inequality, social networks function more as sources of emotional support and collective identity, allowing personal magnetism to maintain its psychological buffering effects [19,20].

Social comparison theory further elucidates this moderating role by emphasizing that individuals constantly evaluate their socio-economic position relative to others [21]. In contexts where inequality perception is low, individuals with high personal magnetism may derive greater psychological benefits from their social interactions, as they experience higher levels of trust and reciprocity within their networks. However, in environments of high inequality perception, these social interactions may become more hierarchical, competitive, and psychologically taxing, weakening the protective function of personal magnetism. The persistent awareness of economic disparities can intensify status anxiety and reinforce the negative effects of unfavorable comparisons, making individuals with high personal magnetism even more susceptible to depression [22].

Taken together, these theoretical perspectives suggest that perceived inequality does not neutralize the relationship between personal magnetism and depression but rather strengthens it. As perceptions of inequality intensify, the social and psychological costs of maintaining high personal magnetism increase, amplifying stress and diminishing the emotional benefits typically associated with strong interpersonal ties [23,24]. This insight highlights the interactive nature of individual social capital and structural economic conditions, revealing that personal magnetism may become a double-edged sword in highly stratified social environments. Thus, this study proposes:

**Hypothesis 2:** Inequality perception strengthens the relationship between personal magnetism and depression.

**The mediating role of community trust.** Community trust, conceptualized as the collective expectation of fairness, reciprocity, and social reliability, serves as a fundamental mechanism through which interpersonal dynamics influence psychological well-being [25]. As societies undergo structural transformations, traditional forms of social embeddedness are increasingly replaced by individualized and transactional relationships, leading to the

fragmentation of social trust networks [26]. This erosion of trust weakens informal social support systems, exacerbating feelings of social isolation and psychological distress, which in turn heighten the risk of depression [27]. Within this evolving social landscape, personal magnetism—an individual's ability to form and maintain strong interpersonal connections—plays a crucial role in rebuilding trust structures. Individuals with high personal magnetism are more likely to facilitate trust-based interactions, enhance social reciprocity, and contribute to the reinforcement of collective norms, thereby strengthening community trust and fostering psychological resilience [4,5].

The mediating role of community trust in the personal magnetism-depression relationship is particularly salient in contexts where mental health stigma is pervasive and access to formal psychological interventions is limited [3]. Extensive research suggests that individuals embedded in high-trust communities report lower levels of stress, anxiety, and depressive symptoms, as trust functions as a critical form of social capital that enhances collective efficacy, promotes mutual aid, and reduces social uncertainty [30]. Trust-based networks foster a sense of belonging and inclusion, enabling individuals to access both emotional and instrumental support, which in turn buffers against psychological distress and mitigates the risk of depression [28]. Given that China's mental health infrastructure remains underdeveloped, informal trust networks often act as functional substitutes for institutional support systems, making community trust a key determinant of mental health outcomes, particularly for individuals who experience social or economic precarity [8,9].

The mediating effect of community trust in the personal magnetism-depression relationship operates through three primary pathways. First, individuals with high personal magnetism engage in frequent reciprocal social interactions, reinforcing perceptions of reliability and trustworthiness, which strengthen social bonds and collective trust norms [29]. Second, highly magnetic individuals often occupy central positions in social networks, acting as bridges that facilitate trust diffusion and foster group cohesion [30]. Third, trust cultivated at the interpersonal level spills over into broader community contexts, reinforcing generalized trust norms that extend beyond immediate social circles, thereby amplifying the protective effects of trust on mental health resilience [31,32]. Thus, this study proposes:

**Hypothesis 3:** Community trust mediates the relationship between personal magnetism and depression.

By positioning community trust as a mediating mechanism, this study provides a conceptual framework for understanding how individual social traits translate into broader structural processes, thereby bridging the micro-macro divide in trust and mental health research. Fig 1 visually represents the proposed research framework.

## Methods

### Data and sample

This study employs data from the 2018 and 2020 waves of the China Family Panel Studies (CFPS), a nationally representative longitudinal survey conducted by the Institute of Social Science Survey at Peking University. The CFPS adopts a stratified multistage probability sampling design, capturing a demographically and geographically diverse sample across 25 provinces, municipalities, and autonomous regions. With data collected from approximately 16,000 households, the CFPS offers a robust empirical basis for analyzing socio-economic and psychological dynamics in contemporary China. The data collection process adhered to rigorous methodological protocols, with trained enumerators conducting computer-assisted personal interviews (CAPI) to enhance data reliability, consistency, and precision.

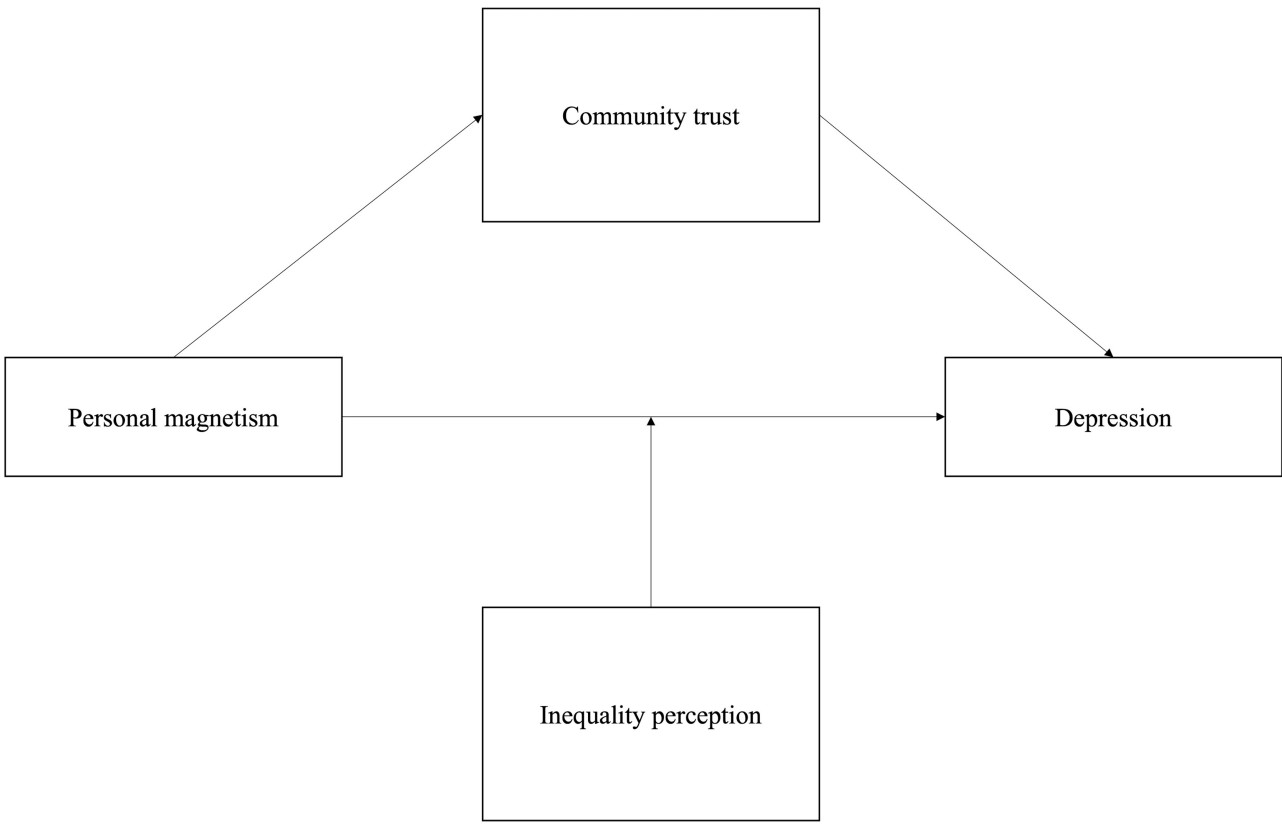

**Fig 1. The conceptual framework of the relationship between personal magnetism and depression.**

To ensure the integrity and robustness of the analytical sample, systematic data preprocessing and quality control procedures were applied. Missing data were identified using diagnostic checks and handled through listwise deletion, retaining only cases with complete information across key study variables. The missing data detection process utilized the egen miss = rmiss(...) function, ensuring that excluded cases did not introduce systematic bias. After applying these exclusion criteria, the final analytical sample consisted of 26,711 respondents, ensuring sufficient statistical power for hypothesis testing.

Ethical approval for this study was obtained from the Peking University Ethics Committee, ensuring adherence to the 1964 Helsinki Declaration and subsequent amendments. The study was conducted in accordance with Articles 38, 39, and 40 of the Constitution of the People's Republic of China, as well as Chapter I, Article 9 of the Statistics Law of the People's Republic of China, which govern the ethical collection and use of national survey data.

Informed consent was obtained verbally from all respondents prior to data collection. Respondents were provided with a detailed explanation of the study's objectives, methodology, potential risks, and data confidentiality procedures. To ensure transparency and procedural integrity, enumerators documented the verbal consent process, including time of agreement, reasons for participation or non-participation, and respondent verification. For respondents under the legal age of 18, informed consent was first obtained from their legal guardians, ensuring that all interviews involving minors were conducted under parental supervision in strict compliance with ethical guidelines for research involving vulnerable populations.

The CFPS adhered to strict ethical standards, and verbal informed consent was obtained from all participants prior to data collection. The process of obtaining verbal consent was documented in detail by the interviewer. Specifically, the interviewer recorded each participant's verbal agreement and confirmation on a consent form, which was then signed by the interviewer as a witness. To further ensure the validity of the process, a second staff member was present during the consent process and also signed the form to act as an additional witness.

## Measures

**Depression.** Depressive symptoms were assessed using a modified version of the Center for Epidemiologic Studies Depression Scale (CES-D), a widely validated instrument demonstrating high reliability and construct validity in the Chinese population [33,34]. The adapted measure in the CFPS consists of six key items, evaluating the frequency of depressive manifestations, including depressed mood, difficulty initiating activities, sleep disturbances, emotional distress, loneliness, and persistent unhappiness. Each item was measured on a four-point Likert scale, ranging from 1 (rarely or never) to 4 (almost daily). The composite score, derived by summing all responses, represents the severity of depressive symptoms, with higher scores indicative of greater psychological distress. Internal consistency reliability was confirmed, with a Cronbach's alpha coefficient of 0.76, suggesting acceptable internal reliability for analytical purposes.

**Personal magnetism.** Personal magnetism was operationalized as an individual's self-perceived social appeal and influence, a construct grounded in social capital and interpersonal attraction theories [35]. The CFPS employed a single-item self-assessment measure, wherein respondents rated their perceived popularity and social influence on a scale from 0 (least magnetic) to 10 (most magnetic). While single-item measures have limitations, prior research suggests that such self-perceptions strongly correlate with broader social capital indicators, making this an efficient proxy for interpersonal influence and embeddedness within social networks [36].

**Inequality perception.** Perceived socioeconomic inequality was measured using a Likert-scale instrument assessing individuals' subjective evaluation of income disparities in their social environment [37]. Respondents rated their perception of inequality on a scale from 0 (very low inequality) to 10 (very high inequality), reflecting the extent to which they believe income disparities affect their society. This measure aligns with the relative deprivation framework, which posits that perceived inequality—beyond objective economic conditions—can significantly shape social attitudes, psychological well-being, and interpersonal trust [21].

**Community trust.** Community trust was conceptualized as individuals' perceived reliability and trustworthiness of others within their local community [38]. The CFPS employed a standardized community trust scale, where respondents rated their trust in neighbors and local institutions on a 0 to 10 scale, with higher scores indicating greater confidence in social cohesion and collective efficacy. This measure aligns with social capital theory, which suggests that higher levels of trust are associated with stronger social networks, increased civic engagement, and enhanced psychological well-being [15].

**Control variables.** Control variables included key demographic, socioeconomic, and behavioral factors based on existing literature on social capital and mental health outcomes [39,40]. These comprised sex (0 = female, 1 = male), age (continuous), marital status (0 = unmarried, 1 = married), education level (1 = below junior high school, 2 = junior high to college, 3 = above college), self-rated health (1 = poor, 5 = excellent), smoking behavior (0 = non-smoker, 1 = smoker), alcohol consumption (0 = non-drinker, 1 = drinker), and household registration (Hukou status, 0 = rural, 1 = urban). The inclusion of these controls accounts for potential biases and ensures robust estimates of the relationships among personal magnetism, community trust, perceived inequality, and depression.

## Statistical analysis

Data analysis was conducted using Stata version 16 (StataCorp, Texas, USA). A multicollinearity diagnostic test using variance inflation factors (VIFs) was performed to assess potential collinearity among predictors, with all VIF values below 2, indicating no substantial multicollinearity concerns. The main analytical approach consisted of multivariate regression models to examine the direct, moderating, and mediating relationships among key study variables [41–43]. First, ordinary least squares (OLS) regression was employed to estimate the direct association between personal magnetism and depression, controlling for relevant covariates. Second, a moderation analysis was conducted by introducing an interaction term (Personal Magnetism × Perceived Inequality) to determine whether the protective effect of personal magnetism on depression varies with inequality perception levels. Third, a mediation analysis was performed within a structural equation modeling (SEM) framework to assess whether community trust mediates the association between personal magnetism and depression. The Bootstrap method (5,000 resamples) was applied to estimate indirect effects, ensuring statistical robustness and bias correction.

To further examine heterogeneity, subgroup analyses by sex were conducted using stratified regression models to assess whether the observed relationships differed between male and female respondents. Additionally, several robustness checks were performed, including alternative model specifications using a fixed-effects ordered logit model and the inclusion of additional control variables (e.g., trust in strangers). These robustness tests confirmed the consistency of the findings across different analytical approaches, reinforcing the validity of the proposed theoretical framework. This comprehensive methodological approach ensures rigorous hypothesis testing and enhances the generalizability of the study's conclusions regarding the interplay between personal magnetism, community trust, perceived inequality, and depression in contemporary China.

## Results

### Descriptive analysis

Table 1 presents the descriptive statistics for the key study variables. The mean depression score across the sample is 13.43 (S. D. = 3.72, range: 8–32), indicating moderate levels of depressive symptoms. Personal magnetism has a mean of 6.99 (S. D. = 1.80, range: 0–10), suggesting that respondents, on average, perceive themselves as moderately socially influential. Inequality perception exhibits a mean of 7.19 (S. D. = 2.27, range: 0–10), highlighting a relatively high awareness of economic disparity within the population. Community trust averages 6.56 (S. D. = 2.04, range: 0–10), reflecting moderate levels of perceived trust in one's community, though with notable individual variation.

Regarding demographic variables, the mean age of respondents is 33.53 years (SD = 11.89, range: 16–96), with the majority falling within the 23–44 age range. A near-equal distribution of sex is observed, with 50.20% of respondents being female and 49.80% male. Marital status distribution indicates that 67.98% of respondents are married, whereas 32.02% remain unmarried. Educational attainment is stratified into three levels, with 23.37% of respondents having an education below junior high school, 65.47% possessing an education level between junior high and college, and 11.16% holding a degree beyond college level.

Health and lifestyle variables provide further context for the sample composition. Self-rated health averages 3.33 (S. D. = 1.10, range: 1–5), with most individuals rating their health as moderate. Smoking behavior is reported by 27.07% of respondents, while 11.35% consume alcohol regularly. A significant rural-urban divide is evident, with 74.71% of respondents residing in rural areas and 25.29% in urban locations. These distributions offer foundational

**Table 1. Descriptive statistics of all variables, CFPS 2018-2020 (N = 26,711).**

| Variable | Mean (SD) | Range | Low | Moderate | High | Category | % (N) |
|---|---|---|---|---|---|---|---|
| Depression | 13.43 (3.72) | 8–32 | 8–12 | 13–19 | 20–32 | | |
| Personal magnetism | 6.99 (1.80) | 0–10 | 0–3 | 4–7 | 8–10 | | |
| Inequality perception | 7.19 (2.27) | 0–10 | 0–3 | 4–7 | 8–10 | | |
| Community trust | 6.56 (2.04) | 0–10 | 0–3 | 4–7 | 8–10 | | |
| Age | 33.53 (11.89) | 16–96 | 16–22 | 23–44 | 45–96 | | |
| Self-rated health | 3.33 (1.10) | 1–5 | 1–2 | 3 | 4–5 | | |
| Sex | | 0–1 | | | | Female | 50.20% (13,409) |
| | | | | | | Male | 49.80% (13,302) |
| Marital status | | 0–1 | | | | Unmarried | 32.02% (8,553) |
| | | | | | | Married | 67.98% (18,158) |
| Education level | | 1–3 | | | | Below Junior High | 23.37% (6,242) |
| | | | | | | Junior High to College | 65.47% (17,487) |
| | | | | | | Above College | 11.16% (2,982) |
| Smoking | | 0–1 | | | | Non-smoker | 72.93% (19,480) |
| | | | | | | Smoker | 27.07% (7,231) |
| Alcohol drinking | | 0–1 | | | | Non-drinker | 88.65% (23,678) |
| | | | | | | Drinker | 11.35% (3,033) |
| Residence registration | | 0–1 | | | | Rural | 74.71% (19,957) |
| | | | | | | Urban | 25.29% (6,754) |

insights into the socio-demographic structure of the sample and its implications for the study's core variables.

## Multivariate regression analysis

Table 2 presents the results of the multivariate regression analysis examining the relationship between personal magnetism, inequality perception, and depression. In Step 1, personal magnetism exhibits a significant negative association with depression ($\beta$ = -0.113, $p < 0.001$), indicating that individuals with higher personal magnetism tend to experience lower levels of depressive symptoms. In Step 2, inequality perception is introduced as an additional predictor and demonstrates a significant positive association with depression ($\beta$ = 0.034, $p < 0.05$), suggesting that individuals perceiving greater economic inequality report higher depressive symptoms.

Step 3 incorporates the interaction term between personal magnetism and inequality perception, which is statistically significant ($\beta$ = -0.016, $p < 0.05$). This finding indicates that the protective effect of personal magnetism on depression is moderated by inequality perception: when inequality perception is lower, the negative association between personal magnetism and depression is stronger, whereas when inequality perception is higher, the protective effect of personal magnetism diminishes. Across all models, self-rated health remains a robust negative predictor of depression ($\beta$ = -0.430, $p < 0.001$ in Step 3), reinforcing its critical role as a determinant of mental well-being.

Table 3 further explores the mediating role of community trust in the relationship between personal magnetism and depression. Personal magnetism continues to exhibit a significant negative association with depression ($\beta$ = -0.113, $p < 0.001$). When community trust is introduced as an additional predictor in Step 2, it demonstrates a negative association with depression ($\beta$ = -0.053, $p < 0.01$), reinforcing the hypothesis that higher levels of trust within one's community buffer against depressive symptoms.

**Table 2. Multivariate regression results integrating inequality perception, CFPS 2018-2020 (N = 26,711).**

| Variable | Step 1 (DV: Depression) | Step 2 (DV: Depression) | Step 3 (DV: Depression) | Step 4 (DV: Personal magnetism) |
|---|---|---|---|---|
| Sex | -1.734 (-1.54) | -1.728 (-1.53) | -1.695 (-1.50) | 0.356 (0.62) |
| Age | 0.034 (1.69) | 0.041* (1.98) | 0.040 (1.95) | -0.018 (-1.73) |
| Marital status | -0.258 (-1.30) | -0.257 (-1.29) | -0.257 (-1.30) | 0.069 (0.68) |
| Education | -0.069 (-0.53) | -0.066 (-0.51) | -0.067 (-0.51) | -0.084 (-1.25) |
| Self-rated health | -0.429*** (-10.88) | -0.429*** (-10.90) | -0.430*** (-10.91) | 0.088*** (4.37) |
| Smoking | 0.181 (1.05) | 0.180 (1.05) | 0.176 (1.02) | -0.036 (-0.41) |
| Alcohol drinking | 0.047 (0.36) | 0.039 (0.29) | 0.039 (0.30) | -0.060 (-0.89) |
| Residence registration | 0.015 (0.09) | 0.015 (0.09) | 0.013 (0.08) | -0.061 (-0.73) |
| Personal magnetism | -0.113*** (-5.07) | -0.116*** (-5.19) | -0.003 (-0.06) | |
| Inequality perception | | 0.034* (2.08) | 0.146** (2.76) | 0.043*** (5.16) |
| Interaction | | | -0.016* (-2.22) | |

t values in parentheses. * p < 0.05, ** p < 0.01, *** p < 0.001. Interaction = Personal magnetism * Inequality perception.

In Step 4, where community trust is modeled as the dependent variable, personal magnetism shows a strong positive association with community trust ($\beta = 0.181$, $p < 0.001$), suggesting that individuals with higher personal magnetism levels are more likely to perceive their communities as trustworthy. Additionally, age is negatively associated with community trust ($\beta = -0.059$, $p < 0.001$), indicating that younger individuals tend to have higher community trust levels. Conversely, better self-rated health is positively associated with community trust ($\beta = 0.067$, $p < 0.01$), suggesting that individuals who perceive themselves as healthier tend to have greater trust in their communities.

## The moderating effect of inequality perception

Table 4 examines how inequality perception moderates the relationship between personal magnetism and depression. The main effect of personal magnetism remains significantly negative ($\beta = -0.118$, $p < 0.001$), reaffirming its protective role against depression. Inequality perception itself is positively associated with depression ($\beta = 0.035$, $p < 0.05$), confirming that individuals with higher awareness of economic disparities are more prone to experiencing depressive symptoms. Crucially, the interaction term between personal magnetism and inequality perception remains statistically significant ($\beta = -0.016$, $p < 0.05$), reinforcing the notion that as inequality perception increases, the protective effect of personal magnetism on depression diminishes.

Fig 2 illustrates this moderating effect, showing that individuals with lower personal magnetism scores tend to report higher depression levels, with the decline in depression being more pronounced for those with lower inequality perception. In contrast, at higher levels of inequality perception, the negative slope of the personal magnetism-depression relationship

**Table 3. Multivariate regression results integrating community trust, CFPS 2018-2020 (N = 26,711).**

| Variable | Step 1 (DV: Depression) | Step 2 (DV: Depression) | Step 3 (DV: Depression) | Step 4 (DV: Community trust) |
|---|---|---|---|---|
| Sex | -1.734 (-1.54) | -1.696 (-1.50) | -1.715 (-1.52) | 0.714 (1.12) |
| Age | 0.034 (1.69) | 0.031 (1.54) | 0.031 (1.52) | -0.059*** (-5.15) |
| Marital status | -0.258 (-1.30) | -0.246 (-1.24) | -0.245 (-1.23) | 0.233* (2.08) |
| Education | -0.069 (-0.53) | -0.070 (-0.53) | -0.068 (-0.52) | -0.011 (-0.14) |
| Self-rated health | -0.429*** (-10.88) | -0.425*** (-10.79) | -0.425*** (-10.79) | 0.067** (3.04) |
| Smoking | 0.181 (1.05) | 0.170 (0.99) | 0.171 (0.99) | -0.210* (-2.17) |
| Alcohol drinking | 0.047 (0.36) | 0.048 (0.36) | 0.047 (0.36) | 0.020 (0.28) |
| Residence registration | 0.015 (0.09) | 0.016 (0.10) | 0.017 (0.10) | 0.017 (0.18) |
| Personal magnetism | -0.113*** (-5.07) | -0.104*** (-4.59) | -0.167** (-2.97) | 0.181*** (14.41) |
| Community trust | | -0.053** (-2.60) | -0.123* (-2.02) | |
| Interaction | | | 0.010 (1.23) | |

t values in parentheses. * $p < 0.05$, ** $p < 0.01$, *** $p < 0.001$. Interaction = Personal magnetism * Community trust.

**Table 4. The moderating role of inequality perception, CFPS 2018-2020 (N = 26,711).**

| Variable | Coef. | Std. Err. | t | p> |t| |
|---|---|---|---|---|
| C_Personal magnetism | -0.118*** | 0.022 | -5.26 | 0.000 |
| C_Inequality perception | 0.035* | 0.016 | 2.11 | 0.035 |
| C_Interaction | -0.016* | 0.007 | -2.22 | 0.026 |
| Sex | -1.695 | 1.127 | -1.50 | 0.133 |
| Age | 0.040 | 0.021 | 1.95 | 0.051 |
| Marital status | -0.257 | 0.198 | -1.30 | 0.195 |
| Education | -0.067 | 0.131 | -0.51 | 0.608 |
| Self-rated health | -0.430*** | 0.039 | -10.91 | 0.000 |
| Smoking | 0.176 | 0.172 | 1.02 | 0.306 |
| Alcohol drinking | 0.039 | 0.131 | 0.30 | 0.766 |
| Residence registration | 0.013 | 0.163 | 0.08 | 0.934 |

* $p < 0.05$, ** $p < 0.01$, *** $p < 0.001$. C_Interaction = C_Personal magnetism * C_Inequality perception.

flattens, indicating that personal magnetism loses some of its protective efficacy in contexts of high perceived economic disparity.

## Sex-based heterogeneity analysis

Table 5 investigates sex-based differences in the effects of personal magnetism, inequality perception, and their interaction on depression. Among females, personal magnetism exhibits a stronger negative association with depression (β = -0.175, p < 0.001), whereas for males, the

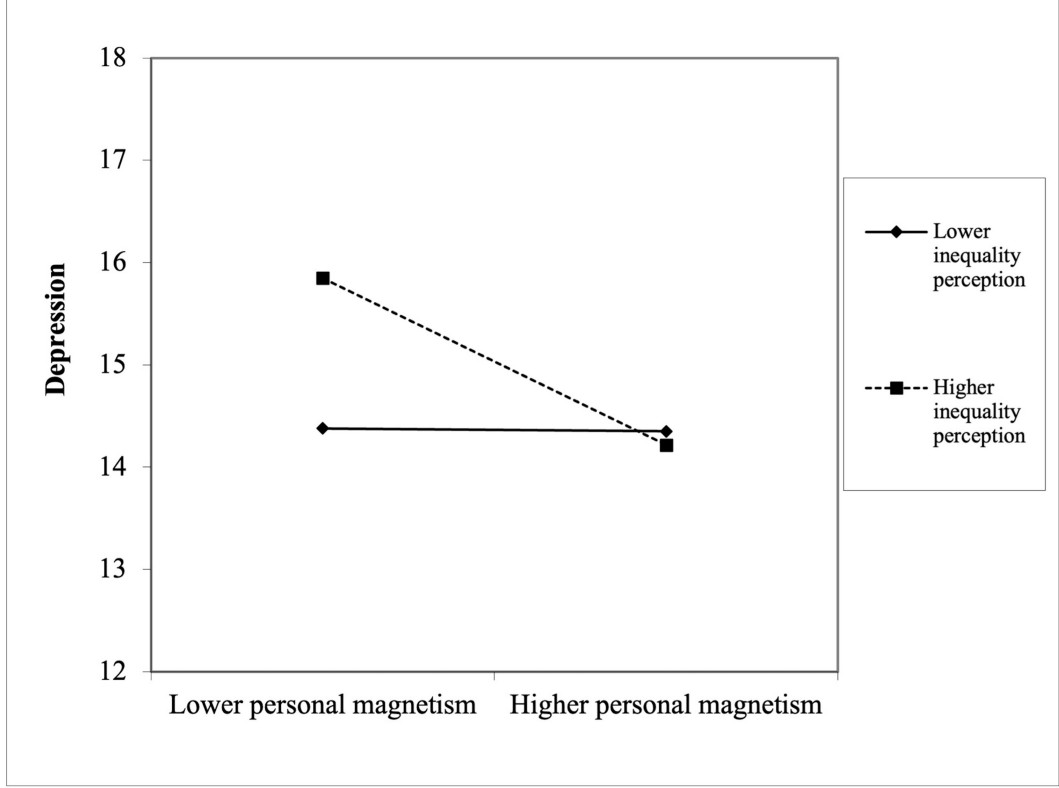

**Fig 2. The moderating role of inequality perception.**

effect is weaker ($\beta$ = -0.063, p < 0.05), suggesting that personal magnetism serves as a more potent protective factor against depression among women. Additionally, inequality perception is positively associated with depression for females ($\beta$ = 0.061, p < 0.05), but this effect is not statistically significant for males, indicating that perceived economic disparities may exert a stronger psychological toll on women.

Furthermore, the interaction term between personal magnetism and inequality perception is non-significant for females ($\beta$ = -0.010, p > 0.05) but remains significant for males ($\beta$ = -0.022, p < 0.05). This suggests that while women experience a direct negative effect of inequality perception on depression, men's mental health outcomes are more contingent upon their level of personal magnetism, particularly in contexts of high inequality perception.

### The mediating effect of community trust

Table 6 evaluates the indirect effect of community trust in mediating the relationship between personal magnetism and depression. The total effect of personal magnetism on depression ($\beta$ = -0.276, p < 0.001) is partially mediated by community trust, as evidenced by the significant indirect effect ($\beta$ = -0.063, p < 0.001). The proportion of the total effect mediated by community trust is 22.9%, indicating that a substantial portion of the protective effect of personal magnetism on depression operates through enhancing trust within one's community.

### Robustness test

To validate the robustness of the findings, several adjustments were made to the sample and model specifications. First, the analysis was re-conducted excluding individuals aged 60 and

**Table 5. Sex-based heterogeneity test results, CFPS 2018-2020 (N = 26,711).**

| Variable | Female | Male |
|---|---|---|
| C_Personal magnetism | -0.175*** (-5.47) | -0.063* (-2.01) |
| C_Inequality perception | 0.061* (2.54) | 0.008 (0.37) |
| C_Interaction | -0.010 (-0.97) | -0.022* (-2.23) |
| Age | 0.038 (1.22) | 0.042 (1.53) |
| Marital status | 0.484 (1.69) | -0.921*** (-3.36) |
| Education | -0.228 (-1.21) | 0.086 (0.47) |
| Self-rated health | -0.488*** (-8.81) | -0.372*** (-6.64) |
| Smoking | 1.456* (2.21) | 0.132 (0.75) |
| Alcohol drinking | 0.179 (0.46) | 0.027 (0.20) |
| Residence registration | -0.073 (-0.34) | 0.172 (0.69) |

* p < 0.05, ** p < 0.01, *** p < 0.001. C_Interaction = C_Personal magnetism * C_Inequality perception.

**Table 6. The mediating role of community trust, CFPS 2018-2020 (N = 26,711).**

| | Effect | Std. Err. | LLCI | ULCI | Ratio |
|---|---|---|---|---|---|
| Total effect (c) | -0.276 | 0.012 | -0.300 | -0.252 | |
| Direct effect (c') | -0.213 | 0.014 | -0.240 | -0.185 | |
| Indirect effect (ab) | -0.063 | 0.004 | -0.071 | -0.056 | |
| Ratio of indirect to total effect mediated | | | | | 0.229 |
| Ratio of indirect to direct effect | | | | | 0.298 |

The "c" path denotes the direct effect of the independent variable on the dependent variable. The "ab" path indicates the indirect effect of the independent variable on the dependent variable through the mediator. Conversely, the 'c' path delineates the effect of the independent variable on the dependent variable when accounting for the mediating variable.

LLCI, Lower Limit of Confidence Interval; ULCI, Upper Limit of Confidence Interval.

above, as mental health assessments may differ for this age group due to factors such as physiological changes, retirement, and social role transitions [44,45]. The original sample included individuals aged 16 to 96 years. The results derived from the modified sample, now limited to individuals aged 16 to 59, were consistent with those from the full sample, thereby reinforcing the robustness of the initial findings.

Second, the fixed-effects regression model was replaced with an ordered logit (ologit) model to account for the ordinal nature of the depression outcome variable [46]. Specifically, the outcome variable was constructed to categorize depression scores into three distinct levels: low (8–12), moderate (13–19), and high (20–32). This categorical classification allowed for a more nuanced analysis of the depression outcomes. This substitution did not alter the substantive conclusions, reinforcing the robustness of the findings across model specifications.

Third, an additional control variable, trust in strangers, was introduced to examine its potential influence on depression, as interpersonal trust is a crucial determinant of social

interactions and mental health outcomes [47]. This variable was measured on a scale ranging from 0 (very low trust) to 10 (very high trust). The incorporation of trust in strangers into the model did not significantly alter the results, further substantiating the robustness of the primary findings. Taken together, these robustness checks confirm the stability of the results across different specifications and sample modifications, thereby enhancing the credibility and generalizability of the conclusions.

## Discussion

This study contributes to the growing body of research on mental health by advancing our understanding of the intricate, multi-layered relationships between personal magnetism, perceived inequality, community trust, and depression within a rapidly transforming socio-economic context. Specifically, it sheds light on how these factors interact to shape mental health outcomes, offering valuable insights into the mechanisms by which social capital and structural inequality converge with individual psychological traits to influence depression. By framing depression as a socially embedded phenomenon, this study moves beyond individualistic explanations and highlights the importance of considering broader socio-structural dynamics.

The findings regarding personal magnetism's role in protecting against depression align with existing theories of social capital, suggesting that individuals with higher social appeal and the ability to forge extensive interpersonal networks experience lower levels of depression [48,49]. However, it is critical to recognize that this protective effect is not solely an outcome of individual charisma or charm, but rather the result of broader social processes that enable individuals to leverage their social capital. In transitional societies like China, where rapid urbanization and market reforms disrupt traditional social structures, personal magnetism serves as a vital tool for navigating the process of social disembedding, which refers to the erosion of stable, long-term social ties [1–3]. By facilitating the formation of new, meaningful relationships, personal magnetism plays a key role in fostering a sense of belonging and stability in environments marked by social flux. This highlights personal magnetism not only as a psychological trait but also as a form of social capital that is especially critical in contexts where formal mental health support structures are underdeveloped.

While personal magnetism serves as a protective buffer against depression, the study also reveals the moderating role of perceived inequality in shaping this dynamic. The interaction between personal magnetism and perceived inequality underscores a paradox that is prevalent in societies experiencing high levels of economic disparity: in such contexts, individuals with greater personal magnetism may find that their social appeal exacerbates feelings of inadequacy [50,51]. This paradox arises from the psychological effects of social comparison processes—where individuals with higher social appeal may engage in upward comparisons that amplify their sense of social and economic deprivation [19,20]. Thus, perceived inequality does not merely affect material conditions; it disrupts the very social networks that could otherwise provide emotional and instrumental support. This calls for a more nuanced conceptualization of social capital, one that accounts for the socio-economic contexts in which it is embedded and recognizes the ways in which inequality may constrain its psychological benefits.

The role of community trust as a mediating mechanism further complicates the relationship between personal magnetism and depression. The study highlights how personal magnetism, by fostering trust, creates a positive feedback loop that reinforces mental health outcomes. Community trust, in this context, functions not just as a facilitator of social cooperation but also as a psychological resource that mitigates depressive symptoms. Trust-based networks offer individuals emotional and practical support, helping to reduce the sense of social alienation that often accompanies depression [52]. This reinforces the notion that

mental health outcomes are shaped by both individual and collective processes: personal magnetism allows individuals to establish trust-based relationships, but it is the broader community context that amplifies or constrains the psychological benefits derived from these networks [53]. As such, community trust acts as a key mediator that strengthens the resilience of individuals, particularly in the face of socio-economic challenges like inequality.

The study's findings have significant implications for both theoretical and practical understandings of mental health. From a theoretical perspective, they highlight the need to rethink the boundaries between individual agency and social structure in mental health discourse. While personal magnetism undoubtedly contributes to psychological resilience, its effects cannot be fully understood outside the context of the structural conditions that shape its expression. The role of perceived inequality in moderating these effects further emphasizes the importance of considering socio-economic factors in mental health research. This suggests that mental health cannot be fully understood in isolation from the broader social and economic structures in which individuals live, thus calling for a more integrated approach to mental health that incorporates both micro- and macro-level factors.

## Conclusion

This study enhances our understanding of mental health by conceptualizing depression as a complex, socially embedded phenomenon, influenced by both individual psychological traits and broader socio-structural factors. By examining the interaction between personal magnetism, community trust, and perceived inequality, the study introduces a multidimensional approach that recognizes the critical role of social structures in shaping mental health outcomes. The findings challenge traditional models of mental health that focus primarily on individual-level factors and emphasize the importance of understanding mental well-being as a dynamic interplay between individual agency and social context.

From a policy perspective, the study highlights the central role of institutions in fostering community trust as a strategy for improving mental health and well-being. Given that community trust mediates the relationship between personal magnetism and mental health outcomes, institutions—especially local governments, non-governmental organizations, and community-based networks—must prioritize the cultivation of trust-based social capital. This involves creating an environment that encourages social cooperation, transparency, and mutual respect, while also addressing the material conditions of inequality through equitable resource distribution. Policies that promote inclusive economic opportunities, universal access to public services, and targeted social welfare programs can help mitigate the adverse effects of inequality on mental health, fostering a more cohesive and resilient society.

Furthermore, policies should aim to strengthen participatory governance structures, where community members are actively engaged in decision-making processes. Such initiatives can foster collective efficacy and social cohesion, further enhancing community trust and reducing the psychological burden of inequality. By addressing both the structural and relational dimensions of social life, these policies can create a more supportive environment for mental health and well-being.

Ultimately, the findings suggest that mental health policy must evolve beyond an individualistic focus to incorporate the social and structural determinants of well-being. By emphasizing community trust and reducing inequality, policymakers can build a more resilient, psychologically stable society. Integrating these social determinants into public health strategies offers a holistic approach to mental health, ensuring that interventions are not only effective but also sustainable in the long term. This study calls for a paradigm shift in mental health policy—one that recognizes the inseparable links between individual mental health, social

capital, and socio-economic conditions, and that prioritizes the creation of trustful, inclusive, and supportive communities.

## Ethical approval

All procedures involving human respondents in this study adhered to the ethical standards of the institutional and/or national research committee, as well as the 1964 Helsinki Declaration and its subsequent amendments or comparable ethical guidelines. The survey was conducted in compliance with Articles 38, 39, and 40 of the Constitution of the People's Republic of China and within the legal framework outlined in Chapter I, Article 9 of the Statistics Law of the People's Republic of China. Ethical approval for this research was granted by Peking University (Ethical Number: IRB00001052-14010).

## Acknowledgment

The author would like to acknowledge the CFPS team for providing the data. The data of CFPS is publicly available at https://doi.org/10.18170/DVN/45LCSO accessed on 25 April 2023.

## Author contributions

**Conceptualization:** Peicheng Wang, Yujie Zhang.

**Data curation:** Peicheng Wang, Yujie Zhang.

**Formal analysis:** Peicheng Wang, Yujie Zhang.

**Funding acquisition:** Yujie Zhang.

**Investigation:** Peicheng Wang, Yujie Zhang.

**Methodology:** Peicheng Wang, Yujie Zhang.

**Project administration:** Peicheng Wang, Yujie Zhang.

**Resources:** Peicheng Wang, Yujie Zhang.

**Software:** Peicheng Wang, Yujie Zhang.

**Supervision:** Peicheng Wang, Yujie Zhang.

**Validation:** Peicheng Wang, Yujie Zhang.

**Visualization:** Peicheng Wang, Yujie Zhang.

**Writing – original draft:** Peicheng Wang, Yujie Zhang.

**Writing – review & editing:** Peicheng Wang, Yujie Zhang.

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
