## [Decision Letter · Decision Letter 0]

9 Feb 2025

PONE-D-24-32639Nesting of groups: Links among personal magnetism, community trust, inequality perception and depressionPLOS ONE

Dear Dr. Zhang,

Thank you for submitting your manuscript to PLOS ONE. After careful consideration, we feel that it has merit but does not fully meet PLOS ONE’s publication criteria as it currently stands. Therefore, we invite you to submit a revised version of the manuscript that addresses the points raised during the review process.

Dear author, please review the comments made, particularly those submitted by the second reviewer, and make the changes following the instructions below. It is necessary that you clearly identify the changes made and their place in the text. Additionally, it is important that you update the literature review regarding interpersonal trust in the context under study. Additionally, it is important that in the first section of the manuscript, just before the end, you justify in a paragraph why the phenomenon investigated in the research is relevant. Finally, in the final section, it is necessary to include a brief section showing the practical implications of the research in relation to public policies. Specifically, it is important to describe the role that institutions play in promoting (social) trust among community members as a strategy to promote the well-being and development of society.

We look forward to receiving your revised manuscript.

Kind regards,

Ignacio Ramos-Vidal, Ph.D.

Academic Editor

PLOS ONE

Journal Requirements:

2. Thank you for stating the following financial disclosure: This study was supported by Chongqing Municipal Social Science Planning Doctoral and Cultivation Project [Grant 2023BS080].

Reviewers' comments:

Reviewer's Responses to Questions

**Comments to the Author**

1. Is the manuscript technically sound, and do the data support the conclusions?

Reviewer #1: Yes

Reviewer #2: Yes

2. Has the statistical analysis been performed appropriately and rigorously? 

Reviewer #1: Yes

Reviewer #2: Yes

3. Have the authors made all data underlying the findings in their manuscript fully available?

Reviewer #1: Yes

Reviewer #2: Yes

4. Is the manuscript presented in an intelligible fashion and written in standard English?

Reviewer #1: Yes

Reviewer #2: Yes

5. Review Comments to the Author

Reviewer #1: It was a well-written article that, while focused on China, can be generalized to other developing and developed countries where communities have become increasingly less cohesive. I was unable to find considerable flaws in the data collection, statistics, or interpretations, and agree with hconclusions.

Reviewer #2: The article is well written with a strong scientific approach. The method used (especially statistical approach) is well described and easily comprehensive in a way that allows to reproduce the analysis done. The data as well as the articles mentionned in the 'References' support the arguments as well as the conclusion.

Few comments have been made and are available in the attached file.

1. In the abstract and the results section, the number of participants stated is 26711. However, on the 18th page, in the Methods section, it is written 26714 participants.

2. Coherence of the article : To improve the flow of the article, I would suggest moving the statistical analysis (below table 1) to a position prior to Table 1, in the Method section.

Additionally, to enhance the coherence of the article, I suggest to introduce Table 1 (descriptive statistics) as the first element in the Results section.

3. Easier comprehension of the descriptive statistics : Include a table or additional text that provides a more detailed breakdown of the ranges for the various key study variables, such as depression, personal magnetism, community trust, and perceived social allure. Specifically, the table or text could outline the score ranges (from 0 to a specified upper limit) for each variable and categorize them into distinct levels (e.g., low, moderate, high). This would offer readers a clearer understanding of how the variables are quantified and help contextualize the results more effectively.

4. Define the age range of the adults included in the study.

6. PLOS authors have the option to publish the peer review history of their article (what does this mean? ). If published, this will include your full peer review and any attached files.

**Do you want your identity to be public for this peer review?** For information about this choice, including consent withdrawal, please see our Privacy Policy .

Reviewer #1: **Yes: ** Kamalakar Surineni

Reviewer #2: No

---

## [Author Response · Author response to Decision Letter 1]

16 Feb 2025

Response to Editor

Point 1: Thank you for submitting your manuscript to PLOS ONE. After careful consideration, we feel that it has merit but does not fully meet PLOS ONE’s publication criteria as it currently stands. Therefore, we invite you to submit a revised version of the manuscript that addresses the points raised during the review process.

Dear author, please review the comments made, particularly those submitted by the second reviewer, and make the changes following the instructions below. It is necessary that you clearly identify the changes made and their place in the text. Additionally, it is important that you update the literature review regarding interpersonal trust in the context under study. Additionally, it is important that in the first section of the manuscript, just before the end, you justify in a paragraph why the phenomenon investigated in the research is relevant. Finally, in the final section, it is necessary to include a brief section showing the practical implications of the research in relation to public policies. Specifically, it is important to describe the role that institutions play in promoting (social) trust among community members as a strategy to promote the well-being and development of society.

• A rebuttal letter that responds to each point raised by the academic editor and reviewer(s). You should upload this letter as a separate file labeled ‘Response to Reviewers’.

• A marked-up copy of your manuscript that highlights changes made to the original version. You should upload this as a separate file labeled ‘Revised Manuscript with Track Changes’.

• An unmarked version of your revised paper without tracked changes. You should upload this as a separate file labeled ‘Manuscript’.

We look forward to receiving your revised manuscript.

Kind regards,

PLOS ONE

Response 1:

Dear Editor,

Thank you for your thoughtful feedback and for providing us with the opportunity to revise our manuscript. We appreciate the valuable insights provided by both the reviewers and yourself. We have carefully addressed all the points raised in your review and made the necessary revisions to enhance the clarity, depth, and overall quality of the manuscript. Below, we provide a detailed response to each of the points mentioned.

1. Justification of the relevance of the phenomenon investigated in the research

In response to your comment about providing a clearer justification for the relevance of the research, we have added a paragraph at the end of the Introduction section. This new addition emphasizes the critical importance of understanding the interplay between personal magnetism, community trust, and inequality perception in the context of rapid socio-economic changes in China. We explain that this study is particularly relevant as China’s transition from a collectivist work-unit system to a more individualized, community-oriented society has led to an erosion of traditional communal networks, contributing to increased feelings of isolation and mental distress. The relevance of our study lies in addressing how personal magnetism can serve as a buffer against these negative mental health outcomes and how community trust plays a crucial mediating role in this process, especially in a society grappling with heightened income inequality.

The revised part is as follows:

Introduction

China’s accession to the World Trade Organization (WTO) marked the onset of an unprecedented economic expansion, catalyzing a profound restructuring of its social fabric [1]. This rapid economic transformation has deepened socioeconomic stratification, eroding communal networks that historically functioned as pillars of social trust and collective identity. As traditional support systems weaken, individuals face diminished access to trust-based social capital, contributing to a fragmentation of interpersonal relationships and a decline in community cohesion [2]. These structural shifts have significant mental health implications, as evidenced by the 6.8% lifetime prevalence of depressive disorders among Chinese adults, alongside persistently low treatment engagement rates, highlighting the urgency of addressing the psychosocial consequences of rapid societal change [3].

By examining the interplay between personal magnetism, community trust, inequality perception, and depression, this study provides critical insights into the mechanisms through which individuals navigate social disembedding and adapt to transforming trust structures. More broadly, this study advances an empirically grounded framework for understanding the role of social trust and interpersonal networks in fostering psychological resilience. Its findings offer policy-relevant strategies for rebuilding social cohesion, reducing mental health disparities, and strengthening collective well-being in societies undergoing profound structural transformation.

2. Literature review regarding interpersonal trust in the context under study

In line with your suggestion, we have significantly updated the Literature Review to provide a more detailed exploration of interpersonal trust, particularly in the context of socio-economic transformations and mental health. This revision draws on recent studies that link community trust with mental well-being, highlighting its role as a critical psychosocial resource in transitional societies like China. We specifically discuss how interpersonal trust networks have been shown to buffer against depressive symptoms, especially when formal mental health services are scarce or stigmatized. We also emphasize that trust not only enhances social cohesion but also supports psychological resilience in individuals who may feel marginalized by societal changes.

The revised part is as follows:

Literature review

Personal magnetism and depression

The construct of personal magnetism occupies a critical position in psychological and social theory, particularly in its role as a protective factor against depressive symptoms. Grounded in social psychology, personal magnetism is conceptualized as an individual’s capacity to attract and maintain social relationships, fostering strong interpersonal connections and social integration. These connections extend beyond superficial interactions, serving as psychosocial resources that reinforce self-worth, provide emotional support, and enhance resilience against adversity [4, 5].

Empirical research has substantiated the protective effects of personal magnetism on mental health, emphasizing the pivotal role of social bonds in mitigating psychological distress. For instance, Kawachi and Berkman assert that the breadth and depth of an individual’s social networks—core components of personal magnetism—buffer against stressors that contribute to depression [6]. Similarly, Cacioppo and Hawkley identify loneliness as the antithesis of social magnetism, demonstrating its strong association with depressive symptomatology, thereby underscoring the importance of social connectedness in mental health regulation [7].

In transitional societies such as China, where rapid socio-economic changes have eroded traditional social structures, personal magnetism assumes an even greater adaptive function. The dissolution of the work-unit (Danwei) system, which historically provided institutionalized social stability through workplace-based communities, has resulted in a fragmentation of trust networks, creating a void that personal magnetism can effectively bridge [8, 9]. This structural transformation necessitates new mechanisms of social integration, positioning personal magnetism as a critical asset in the reconstruction of interpersonal networks and the preservation of psychological well-being [10].

Given that individuals with greater personal magnetism are more likely to establish and sustain meaningful social relationships, which in turn function as buffers against psychological distress [11, 12], this study hypothesizes:

Hypothesis 1: Personal magnetism is negatively correlated with depression.

The moderating role of perceived inequality

The relationship between personal magnetism and depression is not uniform across social contexts but is contingent upon individuals’ perceptions of income inequality. Perceived inequality serves as a moderator, shaping how personal magnetism influences mental health by altering individuals’ social comparisons, expectations of fairness, and access to psychosocial resources. The moderating role of perceived inequality can be understood through the lens of inequity aversion theory, which suggests that individuals who perceive greater economic disparities may experience heightened psychological distress, as such perceptions often generate feelings of social injustice and status anxiety [13, 14]. In this context, even individuals with high personal magnetism may struggle to leverage their social networks as effective protective mechanisms, as widespread perceptions of inequality can erode trust, intensify competition for resources, and amplify the psychological burden associated with social standing [15, 16].

Furthermore, relative deprivation theory provides additional insight into how perceived inequality exacerbates the relationship between personal magnetism and depression. Relative deprivation occurs when individuals feel disadvantaged relative to their peers, fostering frustration, alienation, and lower psychological well-being [17, 18]. Individuals with high personal magnetism often engage in frequent social comparisons, as their social standing is tied to peer validation and interpersonal influence. However, in environments characterized by high perceived inequality, upward social comparisons may become more distressing, leading to diminished self-worth and increased vulnerability to depressive symptoms. In contrast, in societies with lower perceived inequality, social networks function more as sources of emotional support and collective identity, allowing personal magnetism to maintain its psychological buffering effects [19, 20].

Social comparison theory further elucidates this moderating role by emphasizing that individuals constantly evaluate their socio-economic position relative to others [21]. In contexts where inequality perception is low, individuals with high personal magnetism may derive greater psychological benefits from their social interactions, as they experience higher levels of trust and reciprocity within their networks. However, in environments of high inequality perception, these social interactions may become more hierarchical, competitive, and psychologically taxing, weakening the protective function of personal magnetism. The persistent awareness of economic disparities can intensify status anxiety and reinforce the negative effects of unfavorable comparisons, making individuals with high personal magnetism even more susceptible to depression [22].

Taken together, these theoretical perspectives suggest that perceived inequality does not neutralize the relationship between personal magnetism and depression but rather strengthens it. As perceptions of inequality intensify, the social and psychological costs of maintaining high personal magnetism increase, amplifying stress and diminishing the emotional benefits typically associated with strong interpersonal ties [23, 24]. This insight highlights the interactive nature of individual social capital and structural economic conditions, revealing that personal magnetism may become a double-edged sword in highly stratified social environments. Thus, this study proposes:

Hypothesis 2: Inequality perception strengthens the relationship between personal magnetism and depression.

The mediating role of community trust

Community trust, conceptualized as the collective expectation of fairness, reciprocity, and social reliability, serves as a fundamental mechanism through which interpersonal dynamics influence psychological well-being [25]. As societies undergo structural transformations, traditional forms of social embeddedness are increasingly replaced by individualized and transactional relationships, leading to the fragmentation of social trust networks [26]. This erosion of trust weakens informal social support systems, exacerbating feelings of social isolation and psychological distress, which in turn heighten the risk of depression [27]. Within this evolving social landscape, personal magnetism—an individual’s ability to form and maintain strong interpersonal connections—plays a crucial role in rebuilding trust structures. Individuals with high personal magnetism are more likely to facilitate trust-based interactions, enhance social reciprocity, and contribute to the reinforcement of collective norms, thereby strengthening community trust and fostering psychological resilience [4, 5].

The mediating role of community trust in the personal magnetism-depression relationship is particularly salient in contexts where mental health stigma is pervasive and access to formal psychological interventions is limited [3]. Extensive research suggests that individuals embedded in high-trust communities report lower levels of stress, anxiety, and depressive symptoms, as trust functions as a critical form of social capital that enhances collective efficacy, promotes mutual aid, and reduces social uncertainty [30]. Trust-based networks foster a sense of belonging and inclusion, enabling individuals to access both emotional and instrumental support, which in turn buffers against psychological distress and mitigates the risk of depression [28]. Given that China’s mental health infrastructure remains underdeveloped, informal trust networks often act as functional substitutes for institutional support systems, making community trust a key determinant of mental health outcomes, particularly for individuals who experience social or economic precarity [8, 9].

The mediating effect of community trust in the personal magnetism-depression relationship operates through three primary pathways. First, individuals with high personal magnetism engage in frequent reciprocal social interactions, reinforcing perceptions of reliability and trustworthiness, which strengthen social bonds and collective trust norms [29]. Second, highly magnetic individuals often occupy central positions in social networks, acting as bridges that facilitate trust diffusion and foster group cohesion [30]. Third, trust cultivated at the interpersonal level spills over into broader community contexts, reinforcing generalized trust norms that extend beyond immediate social circles, thereby amplifying the protective effects of trust on mental health resilience [31, 32]. Thus, this study proposes:

Hypothesis 3: Community trust mediates the relationship between personal magnetism and depression.

By positioning community trust as a mediating mechanism, this study provides a conceptual framework for understanding how individual social traits translate into broader structural processes, thereb

---

## [Editor Report · Decision Letter 1]

21 Feb 2025

Nesting of groups: Links among personal magnetism, community trust, inequality perception and depression

PONE-D-24-32639R1

Dear Dr. Yujie Zhang,

We’re pleased to inform you that your manuscript has been judged scientifically suitable for publication and will be formally accepted for publication once it meets all outstanding technical requirements.

Kind regards,

Ignacio Ramos-Vidal, Ph.D.

Academic Editor

PLOS ONE

---

## [Editor Report · Acceptance letter]

PONE-D-24-32639R1

PLOS ONE

Dear Dr. Zhang,

I'm pleased to inform you that your manuscript has been deemed suitable for publication in PLOS ONE. Congratulations! Your manuscript is now being handed over to our production team.

Kind regards,

on behalf of

Dr. Ignacio Ramos-Vidal

Academic Editor

PLOS ONE